# IPA (V1): A framework for agent-based modelling of soil water movement

Benjamin Mewes, Andreas H. Schumann

Institute of Hydrology, Water Resource Management and Environmental Engineering, Ruhr University, Bochum, 44801, Germany

*Correspondence to* Benjamin Mewes (Benjamin.Mewes@rub.de)

**Abstract**

In the last decade, agent-based modelling (ABM) became a popular modelling technique in social sciences, medicine, biology and ecology. ABM was designed to simulate systems that are highly dynamic and sensitive to small variations in their composition and their state. As hydrological systems, and natural systems in general, often show dynamic and nonlinear behaviour, ABM can be an appropriate way to model these systems. Nevertheless, only few studies have utilized ABM method for process-based modelling in hydrology. The percolation of water through the unsaturated soil is highly responsive to the current state of the soil system, small variations in composition lead to major changes in the transport system. Hence, we present a new approach for modelling the movement of water through a soil column: autonomous water agents that transport water through the soil while interacting with their environment as well as with other agents under physical laws.

**Keywords**: agent-based modelling, soil water movement, virtual experiment, framework

## 1 Introduction

Agent-based modelling is a relatively new modelling approach or dogma, that was born in social sciences to simulate the interactions of autonomous, encapsulated software agents in a predefined environment that form an emergent system through their interactions and their coupled decision making processes (Macal, North 2010; Jennings 2000; North 2014). Over the years, this technique evolved and came to application in various other disciplines like bio-informatics (Centarowicz et al. 2010), land use modelling (Hammam et al. 2004; Crooks et al. 2008), ecological modelling (Kofler et al. 2014) and policy making (Lempert 2002). In terms of hydrology, the use cases are mostly restricted to watershed management for coupling natural and social systems (Gunkel 2005; Bithell, Brasington 2009; Grashey-Jansen, Timpf 2010; Troy et al. 2015; Bouziotas, Ertsen 2017; Mashhadi Ali et al. 2017; O'Connell 2017). Hydrologic systems are highly dynamic systems which hydrologist try to simulate with a large variety of models from plot-scale to global-scale. Within the modelling community, a manifold of modelling approaches and dogmas exist. Storage models, like HBV, separate the catchment into storages that are connected. These connections are expressed as differential equations to describe the alteration of the storages (Lindström et al. 1997).

More recently, connectivity models like Connectivity of Runoff Models (Reaney et al. 2007; Kirkby 2012) were found to better describe the changing interactions between hydrologically active parts of the catchment or the hillslope in contrast to rather stiff models like HBV. The modelling of physical hydrological systems by ABM is sparse in literature although some approaches were made (Servat 2000; Folino et al. 2006; Parsons, Fonstad 2007; Reaney 2008; Rakotoarisoa et al. 2014; Shao

et al. 2015) but they either relied on a less dynamic predecessor of agent-based modelling, the so-called cellular automata, or were restricted to surface flow models. Generally, Agent-based models allow a deeper analysis of system behaviour, the relation between dynamic components and last but not least, the ability to model unforeseen dynamics in certain model cases that will else be smoothed out in classic numerical models. Hence, the general advantage of Agent-based modelling is to test hypothesis of model component interplay in complex situations with unexpected outcome where known modelling approaches

fail or are hard to parametrize. The unexpected outcome evolves from the multi-agent setup where the model outcome is more than the pure addition of all individual decisions but the result from the interplay of decisions. The system behaviour becomes highly complex as soon as agents have to negotiate the solution of a dilemma situation, like in our case an already saturated target area.  In contrast to purely data-driven approaches, agent-based models rely on clearly formulated rules that create complex behaviour through large numbers of interoperating autonomous software units. We see physical process-based Agent-

based models as appropriate tools for modelling the behaviour of water in the soil matrix where various processes like chemical alteration and physical processes like the adsorption of water in macro pores or the density variation of water through melting-and-freezing processes near permafrost layers are of great hydrologic interest. The modelling of these heterogeneities within the soil, e.g. macro-pores, is an ideal application for agent-based models as here the interplay of dynamic model components, in this case hydrologic agents, plays a significant role due to changes in the interplay of model components and internal states.

Decisions by single agents change not only the model outcome of this very agent, but have a massive influence on the further behaviour of neighbouring agents.  These aforementioned processes show a high level of bi-directionality and connectivity, motivating us to set up a framework with a modelling technique that perfectly fits this emergent and dynamic system behaviour.

In our study we wanted to show the ability of dynamic agent-based models to simulate the movement of water through the

soil. Therefore, we developed an example model within our Integrated Platform for Agent-based modelling framework (IPA) for agent-based hydrological modelling. IPA follows the instructions for an ABM framework setup (North 2014). Our model relies on physical assumptions for percolation speed and distribution of water within the soil matrix. System dynamics are purely based on interactions of hydrologic agents and their environment. With a simple synthetic model setup we showed that a IPA model compares well with other spatially-distributed models, in this case with a model created in the hydrological

framework cmf (Kraft et al. 2011).  We conducted some virtual experiments with cmf to verify the applicability of IPA in order to overcome the need for a real-data test scenario proving the general applicability of this new modelling approach (Weiler, McDonnell 2004). Furthermore, we tested the impact of the type of scheduling and the influence of randomness in the choice of the starting position of hydrologic agents after creation. IPA builds upon the agent-based development environment GAMA (Taillandier et al. 2012; Taillandier et al. 2014) and is published as open source on GitHub.

## 2 Framework development for agent-based modelling of soil water movement and in-soil interactions

Following the requirements of an ABM framework, we introduce the class of the dynamic agents, the class of static agents and the global agent, that controls the modelling experiment as an embracing supervisor (Macal, North 2010; North 2014; Figure 2). In order to set up an ABM for soil water modelling, some principal thoughts on the nature of software agents, their interaction with their environment and eventually the constitution of the model environment have to be done (Crooks et al. 2008). Software agents are encapsulated entities with a defined boundary and attributes, that follow rules to fulfil their goal (Macal, North 2010; Blaschke et al. 2013). Agents interact with their environment through actors and interpret their environment through sensors, whose rules of interaction have to be defined *a-priori* by the modeller (Macal, North 2010; Hofmann et al. 2015). The environment acts in form of a defined number of static agents that comprise all hydrologic agents within their spatial and temporal extent. All actions and interactions are coupled and lead to emergent system dynamics: Agent A decided to perform action I, which hinders Agent B to perform action II but leads B to perform action III and eventually force the environmental layer agent to influence Agent A's future decision. The IPA framework handles all agent classes, the general composition of the modelled system and global model behaviour and is designed to manage agents in a dynamic way to allow the composition of large scale ABM models through the underlying GAMA architecture in headless mode to save computational time ( Taillandier et al. 2012; Boulaire et al. 2015).

In our framework, the global agent manages all static agents (the layers) and dynamic agents (the hydrologic agents).The static agents get information from the hydrologic agents, e.g. how much water is already stored within the layer. Reversely, the layers share information on physical properties to the hydrologic agents. They require this information to calculate their movement speed based on the environmental parameters. Through the knowledge about the hydrologic agents inside the layer, the boundary conditions for each layer are checked.  In contrast to the aforementioned inter-class communication between layer and agents, the intra-class communication of hydrologic agents is crucial for the decision of movement. In dilemma situations, e.g. in case that the target pore space is already covered, the intra-class communication is used to solve that dilemma situation by negotiating the different states of the hydrologic agent. (see Figure 1).

In contrast to classical, equation-based modelling approaches, the amount of parameters for tuning is smaller (van Parunak et al. 1998) but the amount of computational time is higher, which results in a demand for parallelised computation either on GPUs or on high performance systems (Wang et al. 2013). Analysis of ABM results is different to analysis of equation-based models. Generally, the pattern-oriented analysis is to prefer, especially in case of spatially-distributed ABM (Grimm et al. 2005). In our study, we run GAMA in headless mode to save computational time with 2-4 cores for parallel computing of agent states.

## 2.1 Dynamic agents: hydrologic agents

### 2.1.1 Class description of hydrologic agent

Hydrologic agents are carriers of a constant amount of water $w$ that defines their mass (represented as grey circles in Figure 2). All agents carry the same amount of water, but their spatial extent is different because of changing environmental
characteristics. Here, the spatial extent of the hydrologic agents is determined by a circle with radius $r$ that is influenced by the surrounding porosity $\Phi_E$. So, the size of the hydrologic agent may change during its way through the soil column although its mass remains the same, due to a change in the porosity. For future applications the density $\rho$ of the carried water is also included (but set to 1).

$$r = \frac{w}{\Phi_E \rho} \qquad\qquad 1.$$

The influence $I_{hA,L}$ of each hydrologic agent on the static layer agents can be quantified by the area that a hydrologic agent covers of a layer in relation to the complete area of the hydrologic agent (Eq. 2).

$$I_{hA,L} = \frac{A_{hA,L}}{A_{hA}} \qquad\qquad 2.$$

Where $A_{hA,L}$ is the area of layer L covered by Agent hA, and $A_{hA}$ is the area of the hydrologic agent hA. This influence can reach a maximum of 1 if the hydrologic agent covers only one layer, or smaller splits with a sum of 1 per agent, if it covers
multiple layers. This influence is used to calculate the saturation of layers and the surrounding porosity $\Phi E$ of hydrologic agents in the next time step. The saturation of layers $Sat_L$ is calculated by the contributed amount of water $w_{hA}$ of the agents located within the layer ($h_{A,0}$ ... $h_{A,N}$ ) weighted by the influence $I_{hA}$ and the total pore volume of the layer v.

$$Sat_L = \frac{\sum_{hA,0}^{hA,N} I_{hA} \cdot w_{hA}}{V} \qquad\qquad 3.$$

In order to analyse the possible future location of the agent, a cone-shaped view shed is constructed (light grey cones in Figure 2). The view shed has a larger extent than the area of influence, although its length also depends on the radius $r$. Moreover, the saturated percolation speed of the agent in its environment $ks$ and the chosen model time step $\Delta t$ determine the view shed. This can be seen as a tool for numerical integration in the discretised model environment (Servat 2000), as it shows the maximum distance the agent can travel within the next time step. The cone is constructed with an angle of φ =45 ° and the
maximum distance $d$ (Eq. 4) and saturated hydraulic conductivity denoted as ks. The calculation of viewshed is influenced by Darcy's law incorporating the hydraulic conductivity representing the possible step width and the time step of the model. As the agent has a spatial extent, the radius has to be considered as well. The angle φ is a parameter to include the variability of pathfinding due to different grain sizes in the soil structure. As our model setup is 1D soil column and the gradient is limited to one direction, the angle as well is limited to 45° in direction of the gravitational gradient. This angle is chosen because in
the 1D case this angle represents the possible range of direction of movement without a substantially changed gradient. The direction and the speed of movement define the pathfinding algorithm of IPA.

$$d = \sqrt{r \cdot ks \cdot \Delta t} \qquad\qquad 4.$$

### 2.1.2 Rule set for hydrologic agents

The hydrologic agent has to decide whether to move or not to move. Once it has decided whether it shall move, the direction of movement has to be considered. In our case, the rules of movement are defined by physical laws of soil water movement, which can be seen as a trade-off of vertical forces of gravity $\psi_G$ and matrix potential $\psi_M$ that holds water against gravity. These forces are known as the driving potentials. The osmotic potential $\psi_O$ is neglected, which reduces the decision of each agent for movement to:

$$\psi H = \psi M - \psi G \qquad\qquad 5.$$

$\psi H$ equalling 0 means no movement down with the potential gradient takes place, whereas $\psi H > 0$ leads to capillary rise. $\psi H < 0$ results in further deeper percolation of the agent with the speed $k$ of the agent determined by a soil depended retention curve ( van Genuchten 1980;DBG Arbeitsgruppe Kennwerte des Bodengefüges 2009). The speed $k$ is the actual hydraulic conductivity that is higher than the saturated hydraulic conductivity $ks$. In case of an infiltration front moves through a wet soil the matrix potential $\psi M$ can also be in the same sign with the gravitational potential $\psi G$. In case that the future location of the agent at $ti + \Delta t$ is already occupied, the agent tries to find another route following its gradient of potential. Thus the running order, or schedule of hydrologic agents is of importance, which we discuss later in detail. If no other route is possible, the particular agent's movement is suspended for this time step, or tick as it is called in agent-based modelling. The speed of the movement is given by the $k$-value of the surrounding area, which itself depends on the predominant moisture of the environment which is calculated by van Genuchten's model (van Genuchten 1980). This model links soil moisture, the predominant potentials $\psi H$, $\psi M$ and $\psi G$ with the saturated hydraulic conductivity ks and the hydraulic conductivity k which is higher under saturated conditions. The physical soil properties used to calculate van Genuchten's model are given in Tab. 1.

### 2.2 Static agents: Layer agents

### 2.2.1 Class description of layer agent

As stated before, the layers act as static observing agents that survey all dynamic hydrologic agents that belong to their layer (see Figure 2). To each static layer agent a corresponding rectangular area is assigned, later on referenced as the layer. So, the global environment is discretized according to the available data on porosity and layer extents. The total volume of the modelled system is subdivided into a number of single layers. The corresponding soil moisture per layer is calculated by the sum of internal agents' carried water. As the detection of hydrologic affiliation to a specific layer is vulnerable to numerical artefacts from abrupt changes, the calculated soil moisture is smoothed by a univariate spline with a fifth degree. This spline was found to fit the characteristics of soil water content well, but still needs refinement as we show in the detailed analyses. Each layer controls whether the movement of agents is possible, such that problematic situations, e.g. over-saturation of layers,

are avoided. With this the layer is like an internal boundary condition for the decision making process of the hydrologic agents. The interaction between hydrologic agents and the layer agents is bidirectional: So, not only corresponding amounts of water carried by hydrologic agents alter layer processes but also this alteration of soil moisture content is coupled with future agent's decision due to the influence of the soil retention curve on speed and direction of movement (Grashey-Jansen, Timpf 2010).

## 2.2.2 Rule set for layer agents

Static layer agents have various duties: they create hydrologic agents, monitor the soil moisture and oversee that all hydrologic agents act within the boundary conditions. For creation of the hydrologic agents, an infiltration model has to be used. This can be a potential-based agent model, or as well as in this case a Green-Ampt (GA) approach of infiltration leading to the general assumption of a continuous movement of the infiltration front in the matrix. So, the infiltration in the upmost layer represents the upper boundary condition of the model. In this framework an environmental layer can be assigned with a GA infiltration which offers an approximation of GA fairly to compute. (Ali et al. 2016). Ali et al. presented approximation to Green-Ampt where $F(t)$ represents the cumulative infiltration, $S$ the sorption parameter defined by Ali et al. (2016), $ks$ the saturated percolation velocity depending on the soil and $t^*$ is a dimensionless infiltration time (Eq.6). The cumulative infiltration $F(t)$ is transformed into an actual infiltration rate $f(t)$ which sets the number of hydrologic agents at a normally distributed random starting position in the up most layer (Eq. 7). For the calculation of the infiltration into the soil column, the parameters given in Tab.1 are used to calculate the Green-Ampt infiltration in each time step.

$$F(t) = \frac{S^2}{2ks} \left\{ -1 - \left\{ \frac{t^* + \ln\left(1 + t^* + \frac{\sqrt{2t^*}}{1+\frac{\sqrt{2t^*}}{6}}\right)}{\frac{1}{1+t^*+\frac{\sqrt{2t^*}}{1+\frac{\sqrt{2t^*}}{6}}} - 1} \right\} \right\} \qquad 6.$$

$$f(t) = (F(t) - F(t-1))/\Delta t \qquad 7.$$

The mass of the newly generated agents is fixed to a certain amount which limits the maximum number of agents in the system to be simulated. This knowledge is of importance once IPA is able to run on either graphic card accelerated systems or on parallel computing platforms like cloud-based services, because memory allocation and data streaming between processing units becomes the bottleneck of performance and have to be formalized beforehand (Rybacki et al. 2009; Kofler et al. 2014). In contrast to the upper boundary of the model within the IPA framework, the lower boundary is defined by an outflow rate that relies on the $ks$ value of the lowest layer. Once the centroid of the agent, given as the center of the circular shaped agent, has left the system, it dies and the carried amount of water accounts as outflow.

In IPA all layers or agents of the environment, collect information about processes that take place within their extent and along their boundaries. In order to assign a weight depending on the distance of the hydrologic agents to the center of the layer, a density kernel approach is applied to assign weights to each agent to smooth results and reduce numerical and graphical artefacts for the integration of all agent movements that are highly variable and are dependent on the simulated situation.

## 2.3 Scheduling of model actions

Creating agent-based models requires a planned scheme of the running order of processes, actions and actors (e.g. hydrologic agents and global agents) of the model. Thus, the unifying global agent that combines model parameters, hydrologic agents and the observing layer agents, acts as the controlling unit of the whole model (Macal, North 2010). This global agent controls the time and acts as an organizing agent, because it monitors the initialization of the model (at the beginning of the simulation), and asks hydrologic agents to register their layer belonging. Moreover, the global agent is able to force the observing agents to recalculate their state in terms of their current storage.

## 2.4 Model framework for comparison: cmf

For comparison purpose, the cmf framework was used in which a single soil column model was created (Kraft et al. 2011). cmf was chosen because it offers an open framework for spatially-distributed process-based modelling (like solving the Richard's equation for unsaturated flow). Moreover, the general structure of cmf allows a spatial discretization and spatial modelling and can thus be seen as a possible benchmark for a hydrological agent-based modelling framework like IPA. For our model, a single cmf cell, subdivided into 10 layers with a depth of 10cm each, was used. The uppermost layer was connected by a GA infiltration process and a constant head of water available for infiltration upon surface. Transportation of water within the cmf soil column was calculated by the Richards equation for unsaturated flow and Darcy's law for saturated flow. The soil retention curve was modelled with the help of the van-Genuchten-Mualem model. The outlet of the soil column was defined as a free boundary where water can exfiltrate from the system.

## 2.5 Model setup and parametrization of environment

For comparison of the general ability of the usage of IPA for the simulation of soil water movements, a simple synthetic scenario was created. A soil column with a height of 1m and a width of also 1m (leading to a complete volume of 1 m³) was used as model setup. The soil column was divided into 10 single layers with a constant thickness of 10cm. This column was filled with a homogenous sand soil (mS) with soil parameters given in Tab. 1. All parameters applied in the van-Genuchten model influence the calculation of k by the potential gradient and the saturation of the environment, whereas the GA parameters only affect the upper boundary condition. The chosen time step was 1h in order to reduce computational time, keeping in mind that a time step this long is vulnerable to numerical integration problems. By the reduction of time step length, less steps for the hydrologic agents had to be calculated. All layers were equally pre-filled with soil water, resulting in a layer specific $\theta$ of 20 % of available pore space. Although being different in their internal structure, the IPA model and the cmf model (Kraft et al. 2011) shared exactly the same model setup and parametrization. Infiltration was fed by an initial head of water of 1.0m at the surface. The number of ticks was set to 90 which means that both models calculated 90 h or 3.75 days of infiltration and soil water movement. The model time was set long enough (3.75 days) to allow a deeper movement of the infiltration front

through a set of layers, without the head of water on the surface becoming 0. In both models, the time step is chosen as 1h to increase comparability of results and remains constant in all figures and applications.

## 2.6 Performance measures

To estimate the quality of IPA representation of the water movement within the soil column, a suitable measure of performance had to be found. Both models deliver time series of their current states, the layer specific soil moisture θ. Here, we chose to measure the performance of IPA with the r² value.

## 3 Results

Both models showed no numerical errors over the simulation. The volume error of both models in the end with values of about 1% are neglect able. With a runtime of 1.1min on an i5 with 2.2 GHz, 8GB RAM IPA computed only slightly slower than the numerical cmf model that needed about 30s on the same setup. Running IPA in headless mode without graphical output, the computational time was reduced to 48s. Further reduction of computational time could be archived by an outsourcing of the pathfinding to the graphical computation unit. Comparing both model results, it gets obvious that results are not exactly the same, yet the dynamics are similar (Figure 3). The development of soil moisture in the layers follows the same pattern. Saturation reaches a similar level for the first three layers, while the velocity of saturation is different in IPA from the cmf results. Layer 1 does not saturate as fast as in cmf, but movement from Layer 1 to Layer 2 starts earlier in IPA. In cmf, soil water movement from the uppermost layer to the next lower layer starts after approximately 7h while the agent-based model triggers movement of hydrologic agents immediately after the first hour of simulation. After 70 hours both models show saturation in the first layer, so both models reach the same final stable state. In both models the layers are nearly completely saturated at a soil moisture of about 32.8%. The transport from Layer 2 to Layer 3 starts in both models 21h after the beginning of the simulation. Meanwhile, cmf shows a numerical smoother behavior than IPA, while the general system behavior is similar as we can see it in the variation of soil moisture in all layers in IPA, although some numerical oscillations in the soil moisture of Layer 2 become visible.

The express the accordance between IPA and cmf for this run, we calculated the corresponding r² value. Here the mean r² value of the upper three layer scores r² = 0.80. The standard deviation of both models is slightly different 0.039 % (cmf) to 0.045 % (IPA) while the mean values of soil moisture are the same (Table 2).

## 4 Detailed analyses and discussion

### 4.1 Soil column with heterogeneous soil

As stated before, the synthetic case was extended to a more complex situation of two heterogeneous soil types. In order to show the general ability of IPA to model complex systems, the 1D soil column was packed with two different soils leading to the problem of a boundary between two different types of soils with different physical properties. The geometry and the discretization of the grid for the cmf-model remained the same, but the topmost layer consists of Su2 (a weakly silty sand) instead of mS. Su2 was chosen because although it has different physical characteristics, it is still a relative to the original mS soil with a lower share of sand but a higher share of clay. This change of soil type affects highly the process of infiltration and the transition between Layer 1 and Layer 2. None of the other layers was changed, so the ability of IPA to simulate with its pathfinding algorithm (as introduced in Section 2.1.1) and its suspension of movement approach was tested in regard to the added layer transition between Su2 and mS.

Again, both models show a similar, yet slightly different behavior (Figure 4). Transport from Layer 1 to Layer 2 starts immediately as does the movement of water between Layer 2 and Layer 3 in IPA. Saturation of Layer 1 in IPA is reached slower than in cmf but the result after 40h of simulation is a stable system with comparable saturation near full saturation in all layers, although the general IPA behavior is less smooth than cmf. IPA shows slightly higher saturation of about 27% in contrast to 26 % for Layer 2 and 3 in cmf. The saturation of the infiltration Layer 1 shows for both models exactly the same values. The r² value scores at 0.71, which means a high correlation between the outcomes of both models, even though the dynamics between Layer 2 and 3 are different from those in cmf. This could be related to the dilemma of spatiality in the agent-based model as all hydrologic agents have a certain shape and it is likely that this shape has a significant influence on the model outcome. The slightly higher saturation, might be the cause from the boundary conditions that the global surveying agents has to check to avoid oversaturation.

### 4.2 Influence of model scheduling

As mentioned before, scheduling of agent actions is a sensible question in agent-based modelling. Here, three different methods for scheduling were implemented:

- Random calling of agents, that calls agents randomly by chance
- Energy-based scheduling, that allows agents with higher gradients to move first
- Age-based scheduling, allow a movement according to the age (either young first, or old first)

In order to test the influence of the scheduling approaches on the representation capacity of IPA, we conducted a test with the set up presented before, with the same duration of simulation and an identical amount of water available for infiltration on the surface (Figure 5). Random calling of agents is the easiest way to use: Every tick the running order of hydrologic agents is determined randomly. It can be seen that a random scheduling leads to huge smoothing errors because the energy gradient of

each agent (the current state of the agent) is not taken into account. Deeper layers show more fluctuations of soil moisture as it can be seen from tick 80 - 90. To overcome this random approach, an energy-based approach was developed: Those agents with the highest energy-gradient are allowed to move first, which results in smoother results with less numerical fluctuations. This is the case, because the advantage of hydrologic agents with a high potential energy limits conflicts between slow and fast moving agents and like for pathfinding issues through a clearly defined regulation which agents have priority in moving first, trying to get their potentials in balance. Last but not least, we implemented a way to organize the running order by the age of the agents. The age is anticipated by the name, because the unique names of all agents are not reused as soon as an agent leaves the system but originate from consecutive numbering during creation. In our test, old water is allowed to move first, so the scheduling is in a decreasing order. This approach has some problems with the distribution of old water, because the names of those agents that represent old water are rather similar because they were created during initialization.

The correlation coefficients show that all types of scheduling in Layer 1 have less impact than in Layer 2, because all methods do have a high correlation among each other (see Table 3 & Table 4). However, correlation coefficients for Layer 2 show that the energy based approach and the age-based approach have high correlation but struggle less with numerical artefacts like Random Calling in Layer 2. The soil moisture of the upmost layer is in all three cases nearly identical, which shows once more the dependency of the state of the infiltration-affected layer from the chosen infiltration model. From this analysis, it gets clear that the energy-based approach seems to be the best fitting approach. These scheduling approaches may be of interest for upcoming application of IPA because technically this scheduling is the major impact factor on the decision making processes of the hydrologic agents, as one can see in the analysis and can be used for hypothesis testing for the behavior of water.

**4.3 Impact of randomly chosen starting point of hydrologic agents after creation**

Another spatial agent-based modelling specific problem is the starting position for the hydrologic agents within the system. The process of infiltration describes the spatial transition of water from the surface to the soil matrix. Therefore, we assume that each hydrologic agent is located with its complete shape in the topmost layer somewhere near the upper boundary. The x-coordinates within this layer are chosen randomly around the top of the layer, but always as deep in the soil as that its shape is completely inside the layer. In order to verify our assumptions and to show the impact of a different starting positions we show the influence of the chosen starting position for the same model set up with 20 runs. The starting position was chosen by a random normal distribution with $\overline{x} = \overline{W}$ and $\sigma = p \cdot \overline{W}$ where $p$ is varied from 0.1 to 0.9 in 20 steps and $\overline{W}$ is the width of layer, in our case 1m. As one can clearly see, soil moisture in the uppermost layer is only affected by infiltration because calculated soil moisture is nearly constant without any visible influence of the choice of starting position, which makes sense as the hydrologic agent is always located completely within it's the infiltration layer  (see Figure 6). Thus the relevant layers are the deeper layers 2 and 3. Both show slight variations that look like numerical oscillations which makes sense as the smoothing affects the calculation of layer soil moisture because the starting position affects the speed and the pathfinding of the hydrologic agents. The maximum gap of soil moisture per layer is at 3 % for Layer 2 and 3 and at 0.5% at Layer 1 for the

20 runs. Yet, the variance in soil moisture is visible, so, a multi-run of $n$ runs should solve the problem and consequently a mean of these $n$ runs should reduce this numerical artefact effectively that were introduced by the random starting point of the agent (Figure 7).

## 4.4 Weight assignation: From univariate, fitted spline towards more comprehensible methods

In the first step for each hydrologic agent a weight for influence on the layer was assigned by a fitted univariate spline with degree 5 in order to smooth the numerical artefacts from the calculation of the layer affiliation of hydrologic agents. As univariate splines fit well, but interpretation and transfer to other applications is difficult, we have chosen a density kernel estimator with a simple logarithmic distance function to assign a weight, where $w_i$ denotes the weight of the specific hydrologic agent $i$ at distance $d_l$ from the layer $l$ with whom it has a spatial intersection (Eq. 8). This distance is normalized by the

maximum possible distance that a hydrologic agent centroid may have with a corresponding layer agent at distance $ld$, which is defined as the maximum of layer depth or the most far away located agent that still corresponds to the layer's moisture.

$$w_i = \frac{\ln(d_i)}{\max(d_l, ld)} \qquad 8.$$

Hydrologic agents lose their influence on the layer moisture with increasing distance from the static layer agent representing the layer center (see Figure 8). Implementing a new weight assignation in IPA removed the demand for smoothing the soil

moisture per layer. Results for the two-layered synthetic case show, that the approach is promising, yet not fully usable because numerical artefacts still appear (especially in Layer 2), where fluctuations around the correct mean soil moisture for this layer occur with the relative strong variation of about 5 % of soil moisture (Figure 9). Layer 1 is modelled better with less fluctuations, the soil moisture raises faster, maximum soil moisture is as well modelled correctly and shows only little numerical oscillation. In Layer 1 the layer affiliation of each agent is only relevant during the transition from the layer of origin

to the target layer. The overall r² value is lower with 0.62 than for the spline smoothing, but mean moisture is nearly the same (Table 6). The general interpretation that the model shows similar dynamics is supported by an r² value that is higher than 0.5, but yet the standard deviation for both modelling approaches is much higher with 0.042 % regarding 0.0332 % for cmf or 0.0363 % for spline-based IPA. The kernel-based weighting approach looks promising, as the chosen function is easier to interpret than a univariate spline. But, for future applications, this weight determination has to be improved and might be as

well part of a study regarding different distance weighting functions and the construction of a method to quickly find the appropriate function.

## 5 Conclusion and Outlook

It was shown that the relatively new modelling approach of agent-based modelling can be used for answering detailed physical-based hydrological questions like the movement of soil water through a soil column. It was shown, that an agent-based

approach performs well and that its results are comparable to those of a classical Richard's model with Green-Ampt infiltration

set up within the cmf framework which is found to be a suitable environment for modelling complex, spatially distributed hydrological situations physical-based approach. The comparison revealed some further tasks as problems arise from the agent-based modelling dogma: The smoothing for the calculation of layer moisture needs further refinement, as a spline requires too many degrees of freedom for the task of assignation of weights to each single hydrologic agent (Servat 2000). A different

kernel function will be used (instead of a univariate spline) for better explanatory power of the smoothing process that is needed to compare the highly discretized hydrologic agents with the rather rugged layers in cmf. Moreover, the scheduling needs more refinement, especially in terms of age-based scheduling that still has a high random component, as it got clear during the analysis of different scheduling techniques. The computational time of the IPA model is slightly higher than for the cmf model. The required computation time could be further lowered by running the framework headless, which could be a

suitable approach for multi-run optimization approaches like Monte-Carlo.

In future research on this topic we will take a closer look on possible age distributions for hydrologic agents that represent old water. By that, we wish to ameliorate the suspension process that hinders a hydrologic agent from moving in favor of another hydrologic agent that blocks the route along the gradient of forces. So, the commonly observed phenomenon of residual old

water or pushed out old water due to fresh water intrusion can be modelled. In fact, an age-based scheduling also allows finer modelling of hydro-chemical and small-scale pedophysical processes that occur during the transport of water through the soil matrix than common storage models. Another interesting usage of such a refined age-based scheduling is the residence time of water within a coarse rock glacier where melt water is released during rather short melting periods and the water draining from these rock glaciers shows different signatures of age, proving that some water refreezes during the melt and its drain is

delayed to later melting periods. Potential applications of spatially distributed hydrological agent-based models are numerous and IPA might be a suitable framework to answer more complex hydrological questions by adding new rule sets. For the modelling of macro-pore effects on the soil water movement, the principle of agent-based modelling can be interesting: The fastest hydrologic agents wet the surrounding matrix and allow the following water to use the macro-pore as a short-cut through the matrix until the pore is filled. Through the existence of other hydrologic agents the macro-pore is filled and travel speed is

lowered which results in an alternative pathfinding through the matrix because the potential gradient allows a dispersion from the pore to the matrix. We will investigate this application of preferential flow in a hydrological model of a coarse rock glacier.

All in all, one can say that IPA and generally agent-based hydrological process models are at their beginning. In times of big data and a plethora of highly resolved data, this new modelling approach can be of use for those questions where system

behavior can easier be described with dynamic agent-based models than with stiff storage-based models. Within our examples we showed that the new modelling approach is as good (or as bad) as traditional, storage-based models but offer the variability of extension by rules and different scheduling routines. Even at this stage, an agent-based process model offers a great variability in model design for future research questions as it is able to depict the changing dynamics of model components

like in nutrient transport models or complex rock glacier models with changing internal model constellations. By that, the aforementioned variable inner structure of agent-based models extends the modeler's capabilities to describe those systems.

**Code and data availability**

The framework and model are available on GitHub under the following link for general use: https://github.com/HydroMewes/IPA or under DOI: 10.5281/zenodo.1117558. Example data for soil types mS and Su2 is available in the previously mentioned GitHub repository.

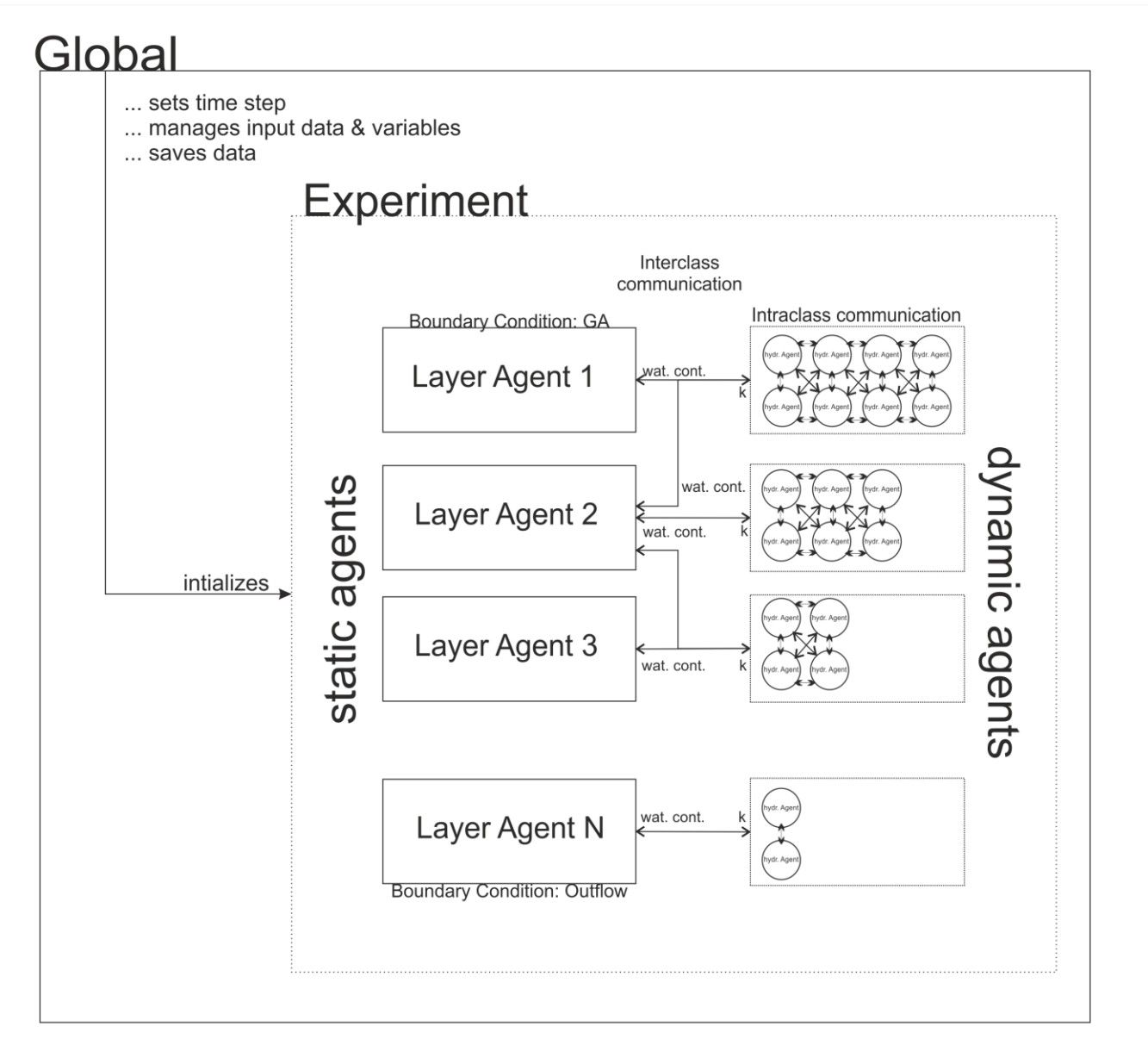

**Figure 1: Conceptual scheme of IPA showing the global agent, the static layer agents and the dynamic hydrologic agents that are glued to an experiment. The intra-class communication between hydrologic agents is needed for the pathfinding of the agents as potential targets for movement might already be occupied. The interclass communication between hydrologic agents and layer agents shows the information exchange from the dynamic agents to the static agents and vice versa. The bi-directional communication between classes is needed to check boundary conditions like the maximum soil moisture per layer and calculate the potential gradient. Moreover it is used to estimate the velocity $k$ of the hydrologic agent according to their surrounding environment.**

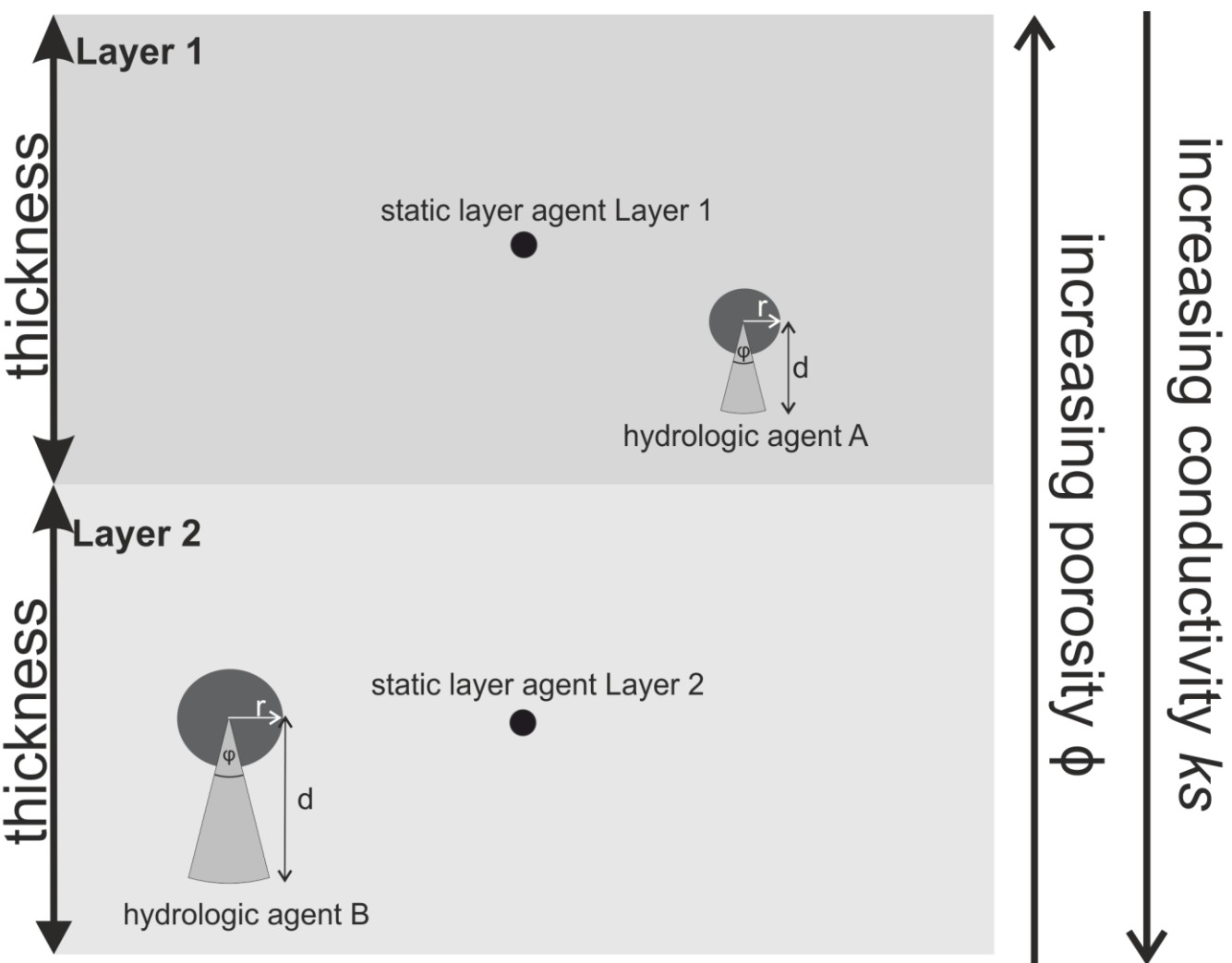

**Figure 2: IPA scheme with two layers with decreasing porosity per depth and two hydrologic agents.**

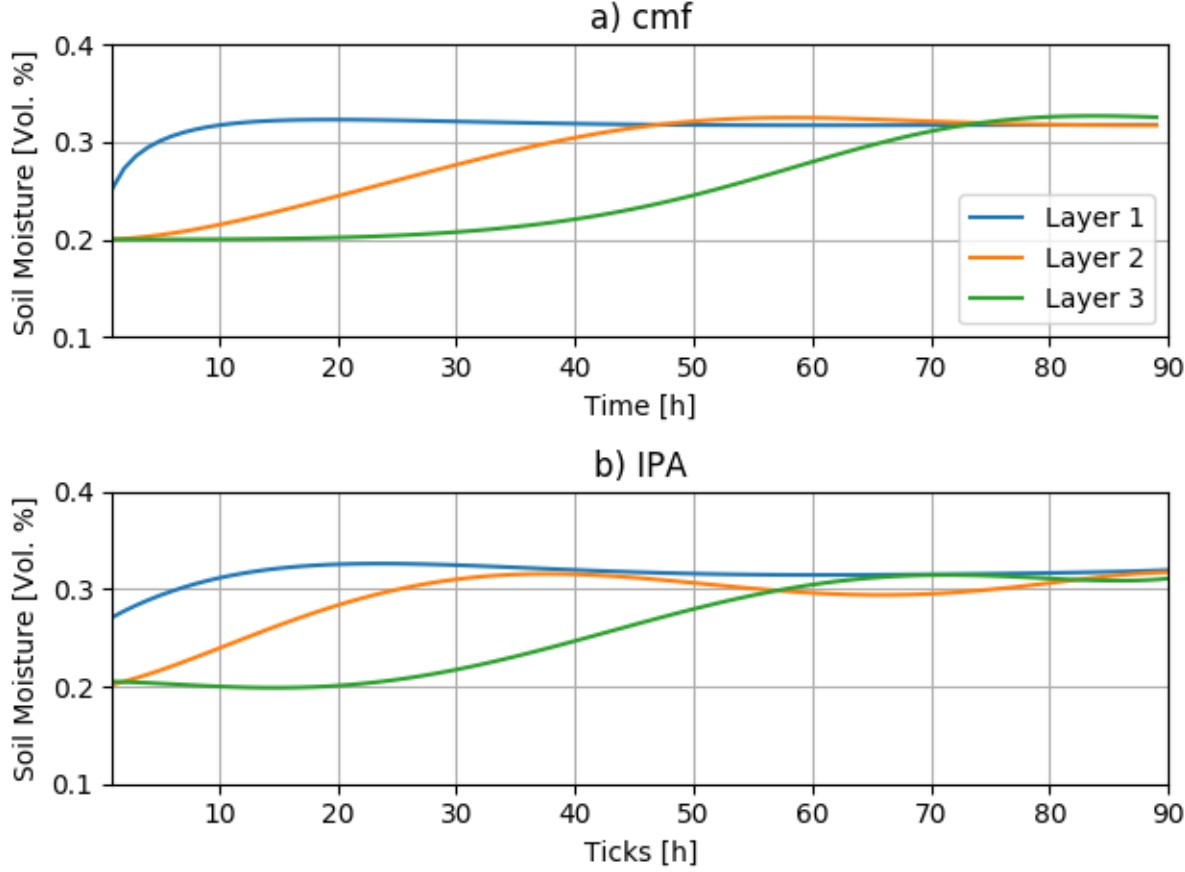

Figure 3: Comparison of soil moisture development of the upmost three layers with a homogenous soil in the column.

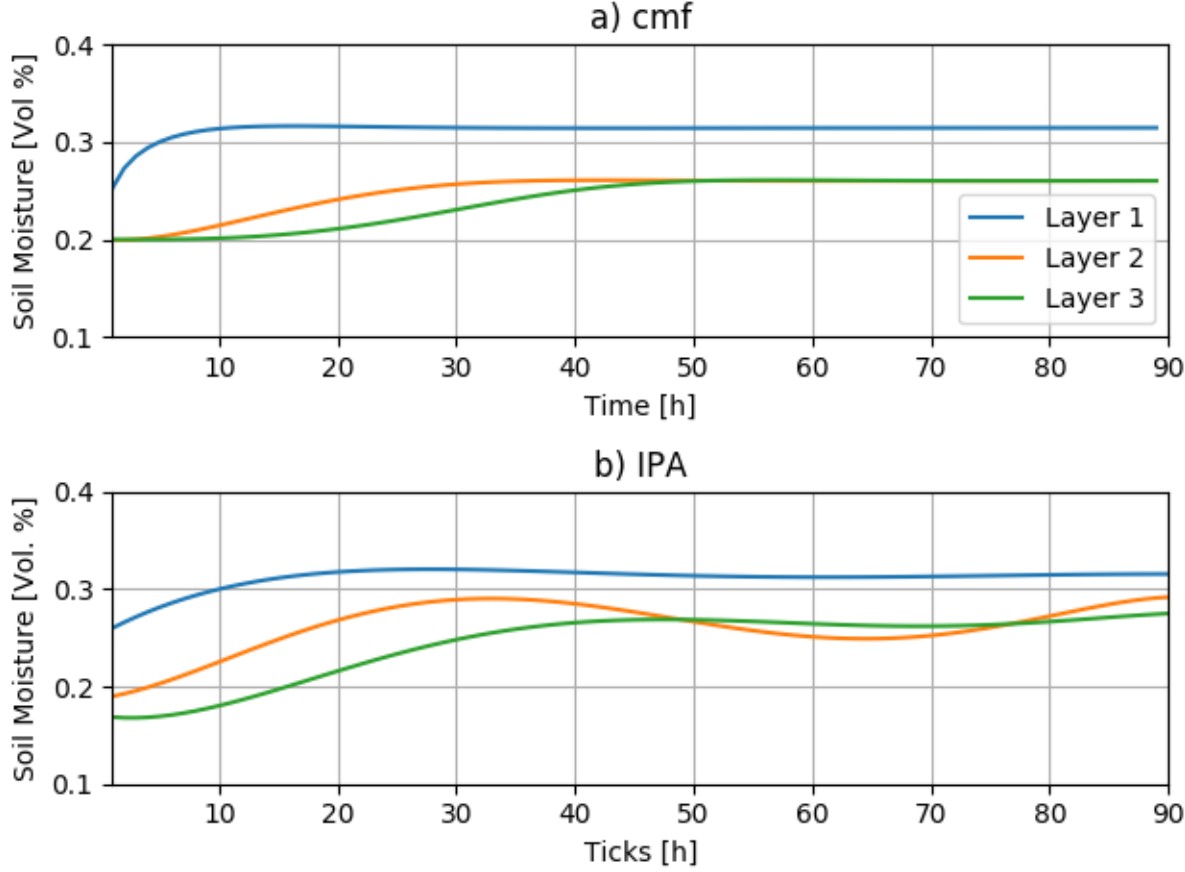

**Figure 4: Comparison of soil moisture development of the upmost three layers with a transition boundary between Layer 1 and Layer 2.**

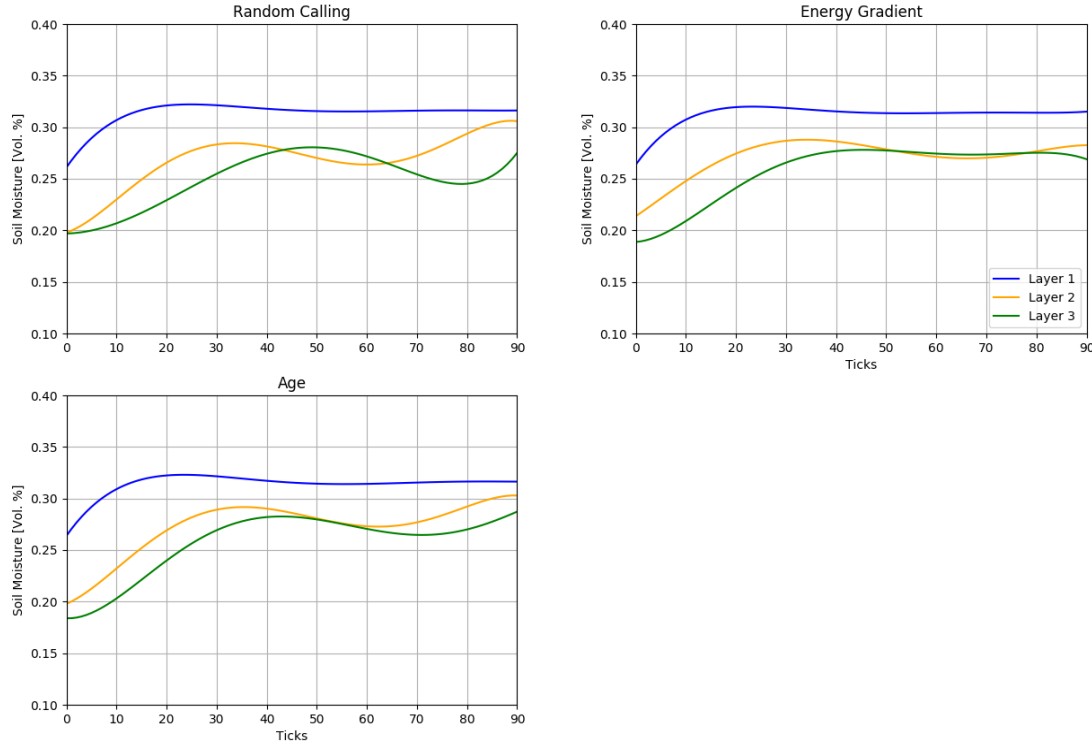

**Figure 5: Analysis of different scheduling methods for soil column with two different soils.**

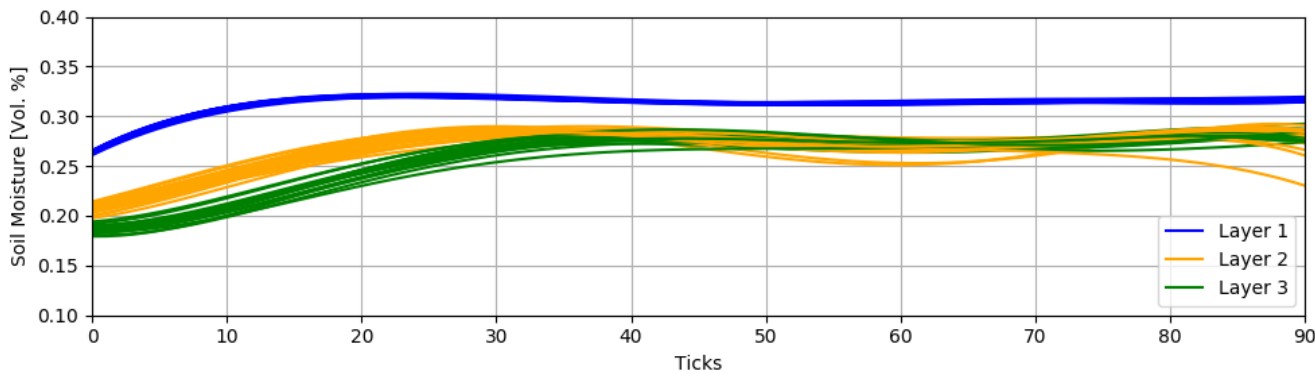

**Figure 6: Influence of randomly chosen starting point. Calculated with 20 runs and a model setup with two different soil types.**

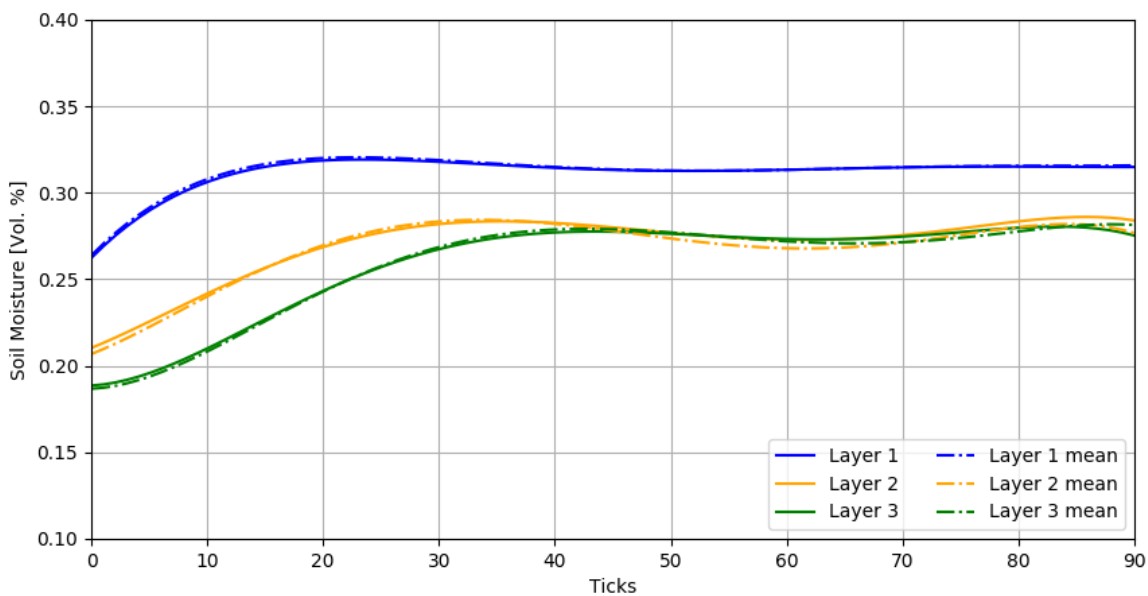

**Figure 7: Mean resulting soil moisture after 20 runs to reduce effects of randomly chosen starting position.**

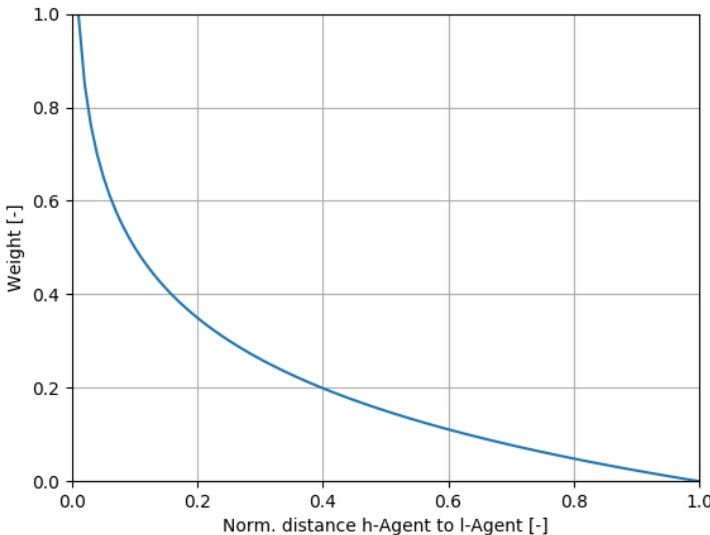

**Figure 8: Decreasing weight with increasing distance of agent's centroid to layer centroid.**

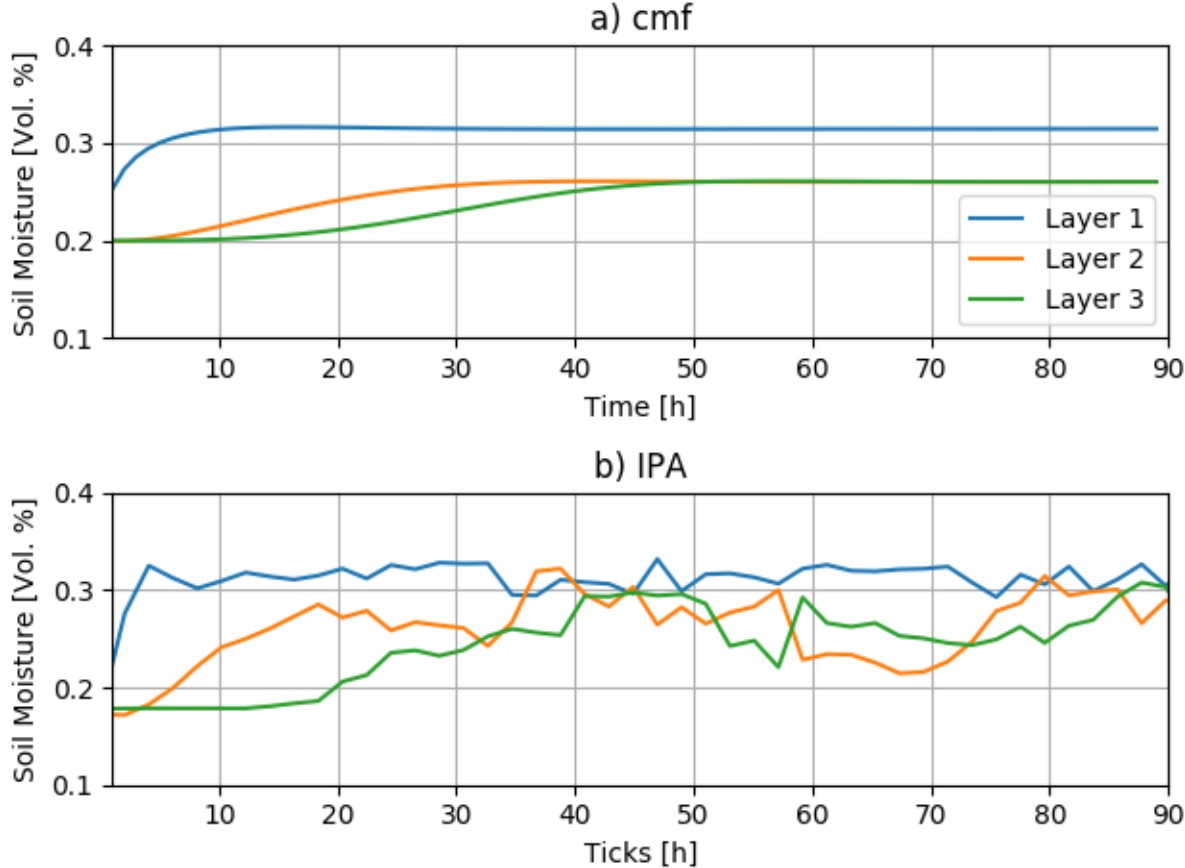

**Figure 9: Modelled soil moisture without spline smoothing but logarithmic kernel weight assignation.**

**Table 1: Physical soil parameters for Green-Ampt model and van Genuchten's model. Source: (DBG Arbeitsgruppe Kennwerte des Bodengefüges 2009).**

| Parameter | Description | Su2 | mS |
|---|---|---|---|
| $Q_r$ [Vol. %] | Residual water content | 0 | 0 |
| $Q_s$ [Vol. %] | Saturated water content | 0.3786 | 0.3886 |
| α [-] | Van-Genuchten-Parameter | 0.20387 | 0.26188 |
| η [-] | Van-Genuchten – Parameter/Green Ampt Parameter | 1.2347 | 1.3533 |
| l [-] | Van-Genuchten -Parameter | -3.339 | -0.579 |
| $k_0$ [mm/d] | Initial hydraulic conductivity | 285.5 | 507.5 |
| $k_s$ [mm/d] | Saturated hydraulic conductivity | 185.0 | 375.0 |

**Table 2: Statistical parameters from model comparison between cmf and IPA for homogenous soil**

| Model | Std [%] | Mean [%] | $r^2$ |
|---|---|---|---|
| cmf | 0.045 | 0.29 | 0.80 |
| IPA | 0.039 | 0.29 | |

**Table 3: Statistical parameters from model comparison between cmf and IPA for inhomogeneous soil**

| Model | Std [%] | Mean [%] | $r^2$ |
|---|---|---|---|
| cmf | 0.033 | 0.27 | 0.71 |
| IPA | 0.036 | 0.28 | |

**Table 4: Correlation coeffecient between scheduling methods for Layer 1**

| | Random Calling | Energy Gradient | Age |
|---|---|---|---|
| **Random Calling** | 1 | 0.73 | 0.53 |
| **Energy Gradient** | 0.73 | 1 | 0.66 |
| **Age** | 0.53 | 0.66 | 1 |

**Table 5: Correlation coefficient between scheduling methods for Layer 2**

| | Random Calling | Energy Gradient | Age |
|---|---|---|---|
| **Random Calling** | 1 | 0.90 | 0.63 |
| **Energy Gradient** | 0.90 | 1 | 0.86 |
| **Age** | 0.63 | 0.86 | 1 |

**Table 6: Statistical parameters from model comparison between cmf and IPA with kernel-based weight determination**

| Model | Std [%] | Mean [%] | $r^2$ |
|---|---|---|---|
| cmf | 0.033 | 0.27 | 0.62 |
| IPA | 0.042 | 0.26 | |

**Publication bibliography**

Ali, Shakir; Islam, Adlul; Mishra, P. K.; Sikka, Alok K. (2016): Green-Ampt approximations: A comprehensive analysis. In *Journal of Hydrology* 535, pp. 340–355. DOI: 10.1016/j.jhydrol.2016.01.065.

Bithell, Mike; Brasington, James (2009): Coupling agent-based models of subsistence farming with individual-based forest models and dynamic models of water distribution. In *Environmental Modelling & Software* 24 (2), pp. 173–190. DOI: 10.1016/j.envsoft.2008.06.016.

Blaschke, Thomas; Hay, Geoffrey J.; Kelly, Maggi; Lang, Stefan; Hofmann, Peter; Addink, Elisabeth et al. (2013): Geographic Object-Based Image Analysis – Towards a new paradigm. In *Isprs Journal of Photogrammetry and Remote Sensing* 87 (100), pp. 180–191. DOI: 10.1016/j.isprsjprs.2013.09.014.

Boulaire, Fanny; Utting, Mark; Drogemuller, Robin (2015): Dynamic agent composition for large-scale agent-based models. In *Complex Adaptive Systems Modeling* 3 (1), p. 1. DOI: 10.1186/s40294-015-0007-2.

Bouziotas, D.; Ertsen, M. (2017): Socio-hydrology from the bottom up: A template for agent-based modeling in irrigation systems. In *Hydrol. Earth Syst. Sci. Discuss.* 2017, pp. 1–27. DOI: 10.5194/hess-2017-107.

Centarowicz, Krzysztof; Paszyński, Maciej; Pardo, David; Bosse, Tibor; La Poutré, Han (2010): Agent-based computing, adaptive algorithms and bio computing. In *ICCS 2010* 1 (1), pp. 1951–1952. DOI: 10.1016/j.procs.2010.04.218.

Crooks, Andrew; Castle, Christian; Batty, Michael (2008): Key challenges in agent-based modelling for geo-spatial simulation. In *GeoComputation: Modeling with spatial agents* 32 (6), pp. 417–430. DOI: 10.1016/j.compenvurbsys.2008.09.004.

DBG Arbeitsgruppe Kennwerte des Bodengefüges (2009): Bodenphysikalische Kennwerte und Berechnungsverfahren für die Praxis / Fachgebiete Bodenkunde, Standortkunde und Bodenschutz, Inst. für Ökologie. With assistance of M. Renger, K. Bohne, M. Facklam, T. Harrach, W. Riek, W. Schäfer, Gerd Wessolek. Berlin: TU Berlin, Selbstverl. (Bodenökologie und Bodengenese, 40).

Folino, Gianluigi; Mendicino, Giuseppe; Senatore, Alfonso; Spezzano, Giandomenico; Straface, Salvatore (2006): A model based on cellular automata for the parallel simulation of 3D unsaturated flow. In *Parallel Computing* 32 (5–6), pp. 357–376. DOI: 10.1016/j.parco.2006.06.003.

Grashey-Jansen, Sven; Timpf, Sabine (2010): Soil Hydrology of Irrigated Orchards and Agent-Based Simulation of a Soil Dependent Precision Irrigation System. In *Advanced Science Letters* 3 (3), pp. 259–272. DOI: 10.1166/asl.2010.1124.

Grimm, Volker; Revilla, Eloy; Berger, Uta; Jeltsch, Florian; Mooij, Wolf M.; Railsback, Steven F. et al. (2005): Pattern-oriented modeling of agent-based complex systems: lessons from ecology. In *Science (New York, N.Y.)* 310 (5750), pp. 987–991. DOI: 10.1126/science.1116681.

Gunkel, Anne (2005): The application of multi-agent systems for water resources research–Possibilities and limits. Master Thesis. Albert-Ludwigs-Universität Freiburg, Freiburg, Germany. Institute of Hydrology.

Hammam, Yasser; Moore, Antoni; Whigham, Peter; Freeman, Claire (Eds.) (2004): Irregular vector-agent based simulation for land-use modelling. 16th Annual Colloquium of the Spatial Information Research Centre. Dunedin, New Zealand.

Hofmann, Peter; Lettmayer, Paul; Blaschke, Thomas; Belgiu, Mariana; Wegenkittl, Stefan; Graf, Roland et al. (2015): Towards a framework for agent-based image analysis of remote-sensing data. In *International Journal of Image and Data Fusion* 6 (2), pp. 115–137. DOI: 10.1080/19479832.2015.1015459.

Jennings, Nicholas R. (2000): On agent-based software engineering. In *Artificial Intelligence* 117 (2), pp. 277–296. DOI: 10.1016/S0004-3702(99)00107-1.

Kirkby, M. J. (2012): Do Not Only Connect. In A. Abbasi, N. Giesen (Eds.): EGU General Assembly Conference Abstracts, vol. 14 (EGU General Assembly Conference Abstracts), p. 3521.

Kofler, Klaus; Davis, Gregory; Gesing, Sandra (2014): SAMPO: an agent-based mosquito point model in OpenCL. In : Proceedings of the 2014 Symposium on Agent Directed Simulation. Tampa, Florida: Society for Computer Simulation International, pp. 1–10.

Kraft, Philipp; Vaché, Kellie B.; Frede, Hans-Georg; Breuer, Lutz (2011): CMF: A Hydrological Programming Language Extension For Integrated Catchment Models. In *Environmental Modelling & Software* 26 (6), pp. 828–830. DOI: 10.1016/j.envsoft.2010.12.009.

Lempert, Robert (2002): Agent-based modeling as organizational and public policy simulators. In *Proceedings of the National Academy of Sciences of the United States of America* 99 (Suppl 3), pp. 7195–7196. DOI: 10.1073/pnas.072079399.

Lindström, Göran; Johansson, Barbro; Persson, Magnus; Gardelin, Marie; Bergström, Sten (1997): Development and test of the distributed HBV-96 hydrological model. In *Journal of Hydrology* 201 (1), pp. 272–288. DOI: 10.1016/S0022-1694(97)00041-3.

Macal, Charles M.; North, Michael J. (2010): Tutorial on agent-based modelling and simulation. In *Journal of Simulation* 4 (3), pp. 151–162. DOI: 10.1057/jos.2010.3.

Mashhadi Ali, Alireza; Shafiee, M. Ehsan; Berglund, Emily Zechman (2017): Agent-based modeling to simulate the dynamics of urban water supply: Climate, population growth, and water shortages. In *Sustainable Cities and Society* 28, pp. 420–434. DOI: 10.1016/j.scs.2016.10.001.

North, Michael J. (2014): A theoretical formalism for analyzing agent-based models. In *Complex Adaptive Systems Modeling* 2 (1), p. 3. DOI: 10.1186/2194-3206-2-3.

O'Connell, Enda (2017): Towards Adaptation of Water Resource Systems to Climatic and Socio-Economic Change. In *Water Resources Management* 31 (10), pp. 2965–2984. DOI: 10.1007/s11269-017-1734-2.

Parsons, Jay A.; Fonstad, Mark A. (2007): A cellular automata model of surface water flow. In *Hydrol. Process.* 21 (16), pp. 2189–2195. DOI: 10.1002/hyp.6587.

Rakotoarisoa, Mahefa Mamy; Fleurant, Cyril; Amiot, Audrey; Ballouche, Aziz; Communal, P. Y.; Jadas-Hécart, Alain et al. (2014): Agents-based modelling for hydrological surface processes on a small watershed (Layon, France). In *International Journal of Geomatics and Spatial Analysis / Revue Internationale de Géomatique* 24 (3), pp. 307–333.

Reaney, S. M. (2008): The use of agent based modelling techniques in hydrology. Determining the spatial and temporal origin of channel flow in semi-arid catchments. In *Earth Surf. Process. Landforms* 33 (2), pp. 317–327. DOI: 10.1002/esp.1540.

Reaney, S. M.; Bracken, L. J.; Kirkby, M. J. (2007): Use of the Connectivity of Runoff Model (CRUM) to investigate the influence of storm characteristics on runoff generation and connectivity in semi-arid areas. In *Hydrological Processes* 21 (7), pp. 894–906. DOI: 10.1002/hyp.6281.

S. Rybacki; J. Himmelspach; A. M. Uhrmacher (2009): Experiments with Single Core, Multi-core, and GPU Based Computation of Cellular Automata. In : Advances in System Simulation, 2009. SIMUL '09. First International Conference on, pp. 62–67.

Servat, David (2000): Modélisation de dynamiques de flux par agents. Application aux processus de ruisellement, infiltration et érosion. Dissertation. Université Pierre et Marie Curie, Paris. Institut de Recherche pour le developpement.

Shao, Qi; Weatherley, Dion; Huang, Longbin; Baumgartl, Thomas (2015): RunCA: A cellular automata model for simulating surface runoff at different scales. In *Journal of Hydrology* 529, Part 3, pp. 816–829. DOI: 10.1016/j.jhydrol.2015.09.003.

Taillandier, Patrick; Grignard, Arnaud; Gaudou, Benoit; Drogoul, Alexis (2014): Des données géographiques à la simulation à base d'agents : application de la plate-forme GAMA. In *Cybergeo : European Journal of Geography*.

Taillandier, Patrick; Vo, Duc-An; Amouroux, Edouard; Drogoul, Alexis (2012): GAMA: A Simulation Platform That Integrates Geographical Information Data, Agent-Based Modeling and Multi-scale Control. In Nirmit Desai, Alan Liu, Michael Winikoff (Eds.): Principles and Practice of Multi-Agent Systems: 13th International Conference, PRIMA 2010, Kolkata, India, November 12-15, 2010, Revised Selected Papers. Berlin, Heidelberg: Springer Berlin Heidelberg, pp. 242–258.

Troy, T. J.; Konar, M.; Srinivasan, V.; Thompson, S. (2015): Moving sociohydrology forward: a synthesis across studies. In *Hydrology and Earth System Sciences* 19 (8), pp. 3667–3679. DOI: 10.5194/hess-19-3667-2015.

van Genuchten, M. Th (1980): A closed-form equation for predicting the hydraulic conductivity of unsaturated soils. In *Soil science society of America journal* 44 (5), pp. 892–898.

van Parunak, H. Dyke; Savit, Robert; Riolo, Rick L. (1998): Agent-based modeling vs equation-based modeling. A case study and users' guide. In *Lecture notes in computer science* 1534, pp. 10–25.

5    Wang, J.; Rubin, N.; Wu, H.; Yalamanchili, S. (2013): Accelerating Simulation of Agent-Based Models on Heterogeneous Architectures. In : Sixth Workshop on General-Purpose Computation on Graphics Processing Units (GPGPU-6).

Weiler, Markus; McDonnell, Jeff (2004): Virtual experiments. A new approach for improving process conceptualization in hillslope hydrology. In *Journal of Hydrology* 285 (1-4), pp. 3–18. DOI: 10.1016/S0022-1694(03)00271-3.