# Peer review of "IPA (V1): A framework for agent-based modelling of soil water movement"

_Geoscientific Model Development, 2017_

## Referee Comment (RC1) · Anonymous Referee #1 · 21 Mar 2018

Review of

**IPA (V1): A framework for agent-based modelling of soil water movement**

by Benjamin Mewes, Andreas H. Schumann
submitted to Geosci. Model Dev. Discuss.
* * *
**Summary and recommendation to the Editor**

This manuscript proposed a framework for agent-based modelling (ABM) of soil water movement. The proposed ABM was applied to simulate the changes of the soil moisture in soil columns with homogeneous and heterogeneous soils. The simulation results were used to compare with the cmf framework in a single soil column. The results show that the agent-based model performs well and that its results are comparable to the classical physical-based approach (cmf).

The traditional numerical modeling for soil movement involves computational-consuming processes. The proposed agent-based modelling framework gives an innovative and interesting simulation approach for soil water movement. Nevertheless, this manuscript does not provide sufficient information for readers, especially without agent-based modelling background, to understand the framework and follow it. In addition, the explanation of the rule set for layer agents should be enhanced as I mentioned in the following detailed comments. Overall, **I would recommend to the Editor acceptance of the manuscript after a revision**. Detailed comments/questions are listed below.

**Comments and questions**

1. Lines 9-10, since the newly developed agent-based model of soil movement is fully based on the known physical assumptions for percolation and water redistribution within the soil matrix, how can this model produce an unexpected outcome where the known modelling approaches fail, such as preferential flow?

2. In Section 2, a conceptual schematic of the definition of dynamic, static, global, and software agents and their interactions should help to understand the IPA framework.

3. In Figure 1, do the layers have a thickness?

4. For dynamic agents, with the same amount of water, the volume of the agent should be inversely proportional to density. Nevertheless, in equation 1, the radius, r, is proportional to density.

5. Page 3, line 30, how to apply equation 2 to IPA? How to calculate the saturation of layers and the surrounding porosity?

6. Page 4, line 11, how to define or calculate the angle, $\phi$?

7. Is the Eq 3 based on Darcy's law? How?

8. Is the ks related to hydraulic conductivity?

9. In equation 4, the matrix potential can be in the same sign with the gravitational potential during the downward infiltration when the soil is dry in front of a downward wetting front.

10. How does the matrix potential link to the soil moisture?

11. Page 4, line 22, $\Phi H < 0$.

12. Page 4 line 22, line 27, page 5 line 23 and page 6 line 4, what are the differences between k, kf, Ks, ks?

13. Page 5, line 12, what is the "observing layer"?

14. Page 5, line 24, it should be "Eq 5".

15. Does Eq 5 represent the upper boundary condition?

16. What is S in Eq. 5

17. Page 6, line 11:,2.4 Scheduling should be 2.3 ....

18. Page 6, line 31: in Table 1, please define Qr, Qs, , , l, k0, and ks and how they used in setting up the agents. Are Qr and Qs residual water content and saturated water content? The unit of Qr, Qs might not be correct.

19. The unit of Soil Moisture in Figures 2 to 6, and 8 should be amended.

20. What are the computational times of the IPA and cmf for running the infiltration in the 1-m soil column?

21. Page 7, line 20-30, what are the differences between the layers 1 to 3 and subdivided into 10 layers with a depth of 10cm each (page6 line 23)? What is the thickness of each layer (Layers 1 to 3)?

22. Page 8, line 14, what is the pathfinding algorithm?

23. Page 9, line 8, how did the energy of the agents be calculated?

24. Figure 2, is the unit of the ticks still hours?

---

## Referee Comment (RC2) · Anonymous Referee #2 · 22 Mar 2018

Review of the Manuscript [Manuscript # GMD-2017-318]: "IPA (V1): A framework for agent-based modelling of soil water movement" By Benjamin Mewes, Andreas H. Schumann General Comments

The manuscript presents a novel model concept by using an agent-based model to describe soil water movement in layered soils. The authors present in a very clear way how agents could be understood and how they can be set into play with each other within a given framework (here: a layered soil). Thus, they rather present a modelling-concept than a specific modelling software. Generally, it is an interesting and intriguing approach to describe soil water movement from a different perspective than other soil water transport models, which surely is of interest to many scientists working in the fields of quantitative water sciences. To the best of my knowledge there is no publication demonstrating such an application in relation to soil water movement, therefore, the approach presented in this paper is novel and will be a valuable contribution to the geo-scientific community. From my perspective as a non-native speaker, the paper is well written and I could follow the model description easily. Nevertheless, with regard to possible applications there are two points I would like to be discussed prior to publication:

- We know that soil water movement is strongly driven by heterogeneities within soils. One keyword would be "macro-pores". The actual discussion turns partly around possible ways to integrate macro-pore- driven soil water processes into the existing modelling frameworks. The proposed agent-based modelling approach would be a good candidate for developing such a representation. In addition, I find that the reasons why scientists should chose the proposed approach over other existing approaches are not satisfying: The proposed advantage of the agent-based model "Agent-based models allow a deeper analysis of system behaviour, the relation between dynamic components and last but not least, the ability to model unforeseen dynamics in certain model cases. (Page 2, line 6)" is not fully convincing. Even simple 1-D approaches could be analyzed with a focus on variable system properties or time-tagged rainfall input. Hence, I recommend to include the problem of predicting the effect of macro-pores into the introduction & discussion section and to propose an idea how the agent-based approach could improve the representation of such lateral soil heterogeneities.

- Another point is the cpu-time, which is needed for the application of the approach. It is not mentioned, how much effort and computer-power is needed to run the presented conceptual representations of a soil. For a possible use of the modelling approach for the simulation of observed time-series of soil-water storages (e.g. lysimeter data), this would be an important decision-criteria during model-selection. The chosen example time-series to analyze and characterize the agent-based model (e.g. 20 runs for the "starting-point-analysis") indicate to me that the time needed to run the model is comparably high. Hence, I think it should be clearly stated if e.g. an uncertainty analysis of model parameters using Monte-Carlo-schemes or other approaches would be possible. Especially for soil-systems, where seasonal dynamics (climate, vegetation) change the system, an advanced analysis of modelling output might be necessary.

- Page 7, eq. 7: $R^2$, as a statistical criteria to evaluate predictive qualities of models is very common and thus needs no extra explanation. The whole paragraph could be deleted.

Overall, I recommend publication after minor revisions.

---

## Author Comment (AC1) · 26 Apr 2018

We thank both referees for their useful suggestions and questions. We addressed each question and changed the manuscript accordingly. The questions the reviewers posed are of great interest and helped us to substantially improve our manuscript and our framework.

We compiled a document with detailed answers in the supplement of this post. For better readability we added the revised manuscript with the referee comment related changes colored in red (referee 1) and green (referee 2).

Please also note the supplement to this comment:

[Figure]

https://www.geosci-model-dev-discuss.net/gmd-2017-318/gmd-2017-318-AC1-supplement.zip

---

## Author Response (AR1)

**Author's response:**

We thank both referees for their useful suggestions and questions. We addressed each question and changed the manuscript accordingly. The questions the reviewers posed are of great interest and helped us to substantially improve our manuscript and our framework. We highlighted changes according to referee 1 in red, and according to referee 2 in green for better readability.

**Referee Comment 1**

"Lines 9-10, since the newly developed agent-based model of soil movement is fully based on the known physical assumptions for percolation and water redistribution within the soil matrix, how can this model produce an unexpected outcome where the known modelling approaches fail, such as preferential flow?"

10 As the agents are autonomous software units, the unexpected model outcome evolves from the interplay of the software units that need to decide how to behave according to their goal, the boundary conditions and their given individual rule set that controls each of the possible actions. In case that a dilemma situation occurs, the agents have to find a solution within the boundary conditions and their portfolio of actions. Hence, the application of well-known physical rules may lead to unexpected outcome as the agents have to find a solution within the given rule set and an agent-based model is more than the pure sum of

15 its components but of all interactions.
Changed: (p.2, l. 10-13)

"In Section 2, a conceptual schematic of the definition of dynamic, static, global, and software agents and their interactions should help to understand the IPA framework."

20 We added a conceptual schematic explaining inter- and intra-class communication including the classes itself. We hope to hereby present the schematics of the modelling framework more clearly.
Changed: added Figure 1 and p.3, l.17-24

"In Figure 1, do the layers have a thickness?"

25 Yes, the layers have a thickness (in our example 10cm). The thickness is added in the schematic overview. We added in the schematic overview the general thickness without any information on the extent.
Changed: Figure 1 became Figure 2 and we added the layer thickness to the schematic overview (p.14)

"For dynamic agents, with the same amount of water, the volume of the agent should be inversely proportional to density.

30 Nevertheless, in equation 1, the radius, r, is proportional to density."
You're totally right on that. We changed the equation in the text and the code, but as the density was set to 1.0 in our example we did not recognize this conceptual error. A change in the source code will be pushed into GitHub with the next update.

Changed: Eq. 1 $\rightarrow$: $r = \frac{w}{\Phi E \rho}$ (p.4, l.9)

"Page 3, line 30, how to apply equation 2 to IPA? How to calculate the saturation of layers and the surrounding porosity?"

We added the calculation of saturation per layer in the paper (Eq.3): The saturation of layers $Sat_L$ is calculated by the contributed amount of water $w_{hA}$ of the agents located within the layer ($h_{A,0}$ ... $h_{A,N}$) weighted by the influence $I_{hA}$ and the total pore volume of the layer $v$.

Changed: p.4. L 16-17, added Eq. 3

"Page 4, line 11, how to define or calculate the angle, φ"

The angle defines the maximum variance for movement within the gradient. In our 1D example the angle is set to 45° where a movement may be considered as possible without changing the gradient substantially. We added a respective passage in the paper explaining the choice of angle in detail.

Changed: p.4, added l.28-30.

"Is the Eq 3 based on Darcy's law? How?"

Yes, Eq 4 (formerly Eq.3) is based on Darcy's law in order to analyze the foreseeable future of the agent on its way through the soil column. Therefore, we include the hydraulic conductivity and the spatial extent of the agent as well as the model time step. The saturated hydraulic conductivity can be exchanged with the actual conductivity. This increases the observed area of the foreseeable future, but also increases the danger of numerical artefacts in estimation of environmental parameters.

Changed: p4, added l.25-26.

"Is the ks related to hydraulic conductivity?"

Yes, ks represents the saturated hydraulic conductivity. We added a definition in the paper. As mentioned before, we need information about the foreseeable future of the agent to adapt to changes in the model environment.

Changed: p.4, l.25.

"In equation 4, the matrix potential can be in the same sign with the gravitational potential during the downward infiltration when the soil is dry in front of a downward wetting front."

You are right, we will include this point in future model revisions. In our simplistic use-case we do not cover this dry pre-condition. This type of wetting front and its link to the matrix potential would need a different behavioral rule set as it was presented here.

"How does the matrix potential link to the soil moisture?"

The matrix potential and the soil moisture can be related through a retention curve. We added a passage to the paper highlighting the link.

Changed: p.5, added l.17-20.

5   "Page 4, line 22, Φ H <0."

We fixed that typo in the revised manuscript.

Changed: Φ H <0, p. 5, l.9.

"Page 4 line 22, line 27, page 5 line 23 and page 6 line 4, what are the differences between k, kf, Ks, ks".

10   Apparently, some typos sneaked into our manuscript. Ks = ks, which is the saturated hydraulic conductivity, whereas k = kf represents the actual hydraulic conductivity which is linked to the current soil moisture of the agent's environment.

Changed: We adapted all occurences of k,kf,Ks and ks and added a description at p.5, l 10-11.

"Page 5, line 12, what is the "observing layer"

15   Every layer of the framework is an observing layer. We removed that confusing information from the revised manuscript as all layers monitor the hydrologic agents inside to calculate their current state.

Changed: We removed the term "observing"

"Page 5, line 24, it should be "Eq 5"".

20   Fixed that typo where automatic numeration went wrong.

Changed: We double checked again the enumeration of equations.

"Does Eq 5 represent the upper boundary condition?"

Yes, it does. We added a clarification in the revised manuscript.

25   Changed: added description p.6, l.7-8.

"What is S in Eq. 5"

S is the dimensionless sorption parameter of the chosen Green-Ampt approach. For further information see Ali et al. 2016.

Changed: added description p.6, l.10.

"Page 6, line 11:,2.4 Scheduling should be 2.3"

We fixed the enumeration in the revised manuscript.

Changed: respective enumeration in the manuscript

"Page 6, line 31: in Table 1, please define Qr, Qs, , , l, k0, and ks and how they used in setting up the agents. Are Qr and Qs residual water content and saturated water content? The unit of Qr, Qs might not be correct."

We added a detailed explanation of the parameters in Tab. 1. The parameters mainly control the boundary condition for the creation of agents in the Green-Ampt approach. Moreover, they are utilized in the van-Genuchten model for the estimation of

5 movement velocity. As to the unit, of Qr and Qs: The parameters are given as volumetric percent. We clarified this in the text but as we did not measure the parameters on our own, we refer to the literature (Arbeitsgruppe Bodengefüge).

Changed: added description of parameters given in Tab.1, p.19

"The unit of Soil Moisture in Figures 2 to 6, and 8 should be amended."

10 We added the unit (vol. percent) in all figures in the revised manuscript.

Changed: Units of soil moisture add in all figures where missing.

"What are the computational times of the IPA and cmf for running the infiltration in the 1-m soil column?"

We added a passage in the results section dealing with the run time of both modelling approaches: With a runtime of 1.1min

15 on an i5 with 2.2 GHz, 8GB RAM it computed only slightly slower than the numerical cmf model that needed about 30s on the same setup. Running IPA in headless mode without graphical output, the computational time was reduced to 48s. Further reduction of computational time could be archived by an outsourcing of the pathfinding to the graphical computation unit.

Changed: added a passage on runtime p.8, l.7-10.

20 "Page 7, line 20-30, what are the differences between the layers 1 to 3 and subdivided into 10 layers with a depth of 10cm each (page6 line 23)? What is the thickness of each layer (Layers 1 to 3)?"

Each layer has a thickness of 10cm. The main difference is that the upmost layer represents the upper boundary condition where the agents are created. As their starting position is partly random, the upmost layer has a high influence on the shape and the initial velocity of the wetting front and thus on the model performance.

25 Changed: We added a sentence on the layers with their regarding thickness. P.7, l.11.

"Page 8, line 14, what is the pathfinding algorithm?"

The pathfinding follows the gradient and checks the target position whether the space is free or occupied. We clarified the link to the respective position in the text.

30 Changed: P.9, added a link to the pathfinding in section 2.1.1 and added a claryfing line on p.4,l.31-32.

"Page 9, line 8, how did the energy of the agents be calculated?"

The energy of the agents is calculated by the potential energy. We pointed that out in the revised manuscript.

Changed: added potential energy p.10, l.4

"Figure 2, is the unit of the ticks still hours?"

The unit of ticks is constant in the whole model setup and through all figures. We clarified that by a sentence added on p.8, l.1-2: "In both models, the time step is chosen as 1h to increase comparability of results and remains constant in all figures and applications."

Changed: sentence added on p.8, l.1-2

**Referee Comment 2**

"We know that soil water movement is strongly driven by heterogeneities within soils. One keyword would be "macro-pores". The actual discussion turns partly around possible ways to integrate macro-pore- driven soil water processes into the existing modelling frameworks. The proposed agent-based modelling approach would be a good candidate for developing such a

5 representation. In addition, I find that the reasons why scientists should chose the proposed approach over other existing approaches are not satisfying: The proposed advantage of the agent-based model "Agent-based models allow a deeper analysis of system behaviour, the relation between dynamic components and last but not least, the ability to model unforeseen dynamics in certain model cases. (Page 2, line 6)" is not fully convincing. Even simple 1-D approaches could be analyzed with a focus on variable system properties or time-tagged rainfall input. Hence, I recommend to include the problem of predicting the effect

10 of macro-pores into the introduction & discussion section and to propose an idea how the agent-based approach could improve the representation of such lateral soil heterogeneities"

Agent-based modelling is a suitable approach for modelling heterogenetic environmental problems like macro-pore driven water transports. We point out use cases, e.g. macro-pore dominated soils and the infiltration into coarse rock glaciers in the discussion. As we prepare further research on the applications mentioned we did not want to stress these ideas in this

15 fundamental paper. But we agree with you that in the non-revised state the argumentation why this very approach shall be used was too short. Hence, we extended the respective parts in our introduction and discussion.

Changed:

        p. 2, l.17-21, added

        p.12, l.21-26,added

"Another point is the cpu-time, which is needed for the application of the approach. It is not mentioned, how much effort and computer-power is needed to run the presented conceptual representations of a soil. For a possible use of the modelling approach for the simulation of observed time-series of soil-water storages (e.g. lysimeter data), this would be an important decision-criteria during model-selection. The chosen example time-series to analyze and characterize the agent-based model

25 (e.g. 20 runs for the "starting-point-analysis") indicate to me that the time needed to run the model is comparably high. Hence, I think it should be clearly stated if e.g. an uncertainty analysis of model parameters using Monte-Carlo-schemes or other approaches would be possible. Especially for soil-systems, where seasonal dynamics (climate, vegetation) change the system, an advanced analysis of modelling output might be necessary."

We see that computational time might be a limiting factor. We therefore added a passage on the computational time needed

30 for running IPA and cmf. Although computational time is higher for IPA than cmf, the additional time for the agent-based approach is not prohibitive and could be further reduced by computation on graphical processing units and headless mode. We limited our results to 20 runs in the analysis because our preliminary results showed that the expected outcome was rather invariant with 1000 runs. Therefore, we limited the runs as we did not expect higher variance in results. Nevertheless, we saved 30s per run which saved us hours of computational time.

Changed:

        p.8, l.9-12, added.

"Page 7, eq. 7: R2 as a statistical criteria to evaluate predictive qualities of models is very common and thus needs no extra explanation. The whole paragraph could be deleted."

5   Changed: We removed this part of the paper according to your suggestion.

[revised manuscript text omitted]

---

## Author Response (AR2)

Topical Editors comment:

Dear Authors,

Your revised manuscript and responses to the reviewers are satisfactory to GMD's requirement. However, I have one minor suggestion that in your last sentence in the conclusion: "Even at this stage, an agent-based process model offers a great variability in model design for future research questions."

Can you be more specific on what kind of future research? Let me know if you have any questions.

Best,
Topical Editor,
Min-Hui Lo

Authors response:

Dear Mr. Min-Hui Lo,

Thank you for your suggestion. We extended the sentence by examples and hopefully we made it clearer! Our future research on this topic is focused on model with highly dynamic internal structures, like nutrient transport and irrigation modelling. Therefore, we think that ABMs are suitable tools as the internal connections are less restricted than in "classical" numerical models.

On behalf of the authors,

Benjamin Mewes